# Lasting lockdown love? Problem behaviour and pandemic and non-pandemic related risk factors influencing the owner-dog relationship in a UK cohort of dogs reaching early adulthood

**Bree L. Merritt**[1], **Dan G. O'Neill**[2], **Claire L. Brand**[1], **Zoe Belshaw**[3], **Fiona C. Dale**[1], **Camilla L. Pegram**[2], **Rowena M. A. Packer**[1]*

1 Department of Clinical Science and Services, The Royal Veterinary College, Hatfield, Herts, United Kingdom, 2 Department of Pathobiology and Population Sciences, The Royal Veterinary College, Hatfield, Herts, United Kingdom, 3 EviVet Evidence-based Veterinary Consultancy, Nottingham, United Kingdom

\* rpacker@rvc.ac.uk

**Data Availability Statement:** The dataset associated with this publication are available via

## Abstract

The owner-dog relationship is a critical component of sustained dog ownership. Increased understanding of risk factors for weak owner-dog relationships can identify owner-dog dyads at higher risk of poor welfare outcomes, including dog relinquishment and euthanasia. The internationally documented boom in puppy acquisition during the COVID-19 pandemic led to welfare concerns for this cohort of dogs, including impulsive purchasing of puppies to unsuitable homes, increased supply of puppies from poor-welfare sources, and deficits in early-life experiences for puppies. Combined, these changes were feared to lead to problem behaviours, weak owner-dog relationships and increased future relinquishment in this uniquely vulnerable generation. The Pandemic Puppies project longitudinally studied dogs bought as puppies aged < 16-weeks old during the 2020 phase of the COVID-19 pandemic by collecting owner-completed data during puppyhood and as adults. This study aimed to investigate factors associated with the owner-dog relationship in early-adulthood via a cross-sectional analysis of a subset of Pandemic Puppies data ($n$ = 794). When dogs were 21-months old owners completed the Monash Dog-Owner Relationship Scale (MDORS), from which the Perceived Emotional Closeness (Closeness) and Perceived Costs subscales were established as reliable in this sample and were used as outcome variables in multivariable analyses to explore risk factors such as health, behaviour, and acquisition-related variables. Problem behaviours, including those related to lack of control, fear, separation, and aggression were the predominant risk factors associated with increased Perceived Costs score. The presence of most problem behaviours was not associated with reduced Closeness, suggesting a potential source of emotional conflict for owners. However, owners of dogs showing aggressive behaviours had lower Closeness scores. Puppy acquisitions explicitly motivated by the pandemic were associated with increased Perceived Costs. Support interventions

Figshare, at: https://figshare.com/s/
d1854b45cd13d482fdf1?file=49987971.

**Funding:** This research was funded by Battersea
Dogs & Cats Home, www.battersea.org.uk, grant
number BAT21M/Research/002 awarded to R.M.A.
P. as primary investigator and D.G.O. and Z.B. and
co-investigators. The funders had no role in study
design, data collection and analysis, decision to
publish, or preparation of the manuscript.

**Competing interests:** The authors have declared
that no competing interests exist.

targeted at owners of dogs with problem behaviours are of high importance if positive
owner-dog relationships are to be maintained.

## Introduction

The emotional relationship between an owner and their dog is widely recognised as a critical
element of dog ownership with the potential to influence welfare on both the dog and human
sides of the relationship, positively and negatively. A person's welfare is their health and happi-
ness [1], and similarly animal welfare has been defined as how the animal is feeling [2]. Animal
welfare is often interpreted using the Five Domains Model, considering the animal's mental
state regarding their experiences of nutrition, their environment, physical health, and beha-
vioural interactions [3]. A strong owner-dog relationship has been associated with improved
physical health both for the dog [4] and owner [5, 6]. However, dog ownership has been associ-
ated with poorer owner wellbeing across a range of common circumstances including the neg-
ative physical and emotional effects of caregiving for dogs with health or behavioural problems
('caregiver burden') [7, 8] or disenfranchised guilt related to work-family conflict [9]. Weak
owner-dog relationships are associated with a higher risk of dog relinquishment [10]. Relin-
quishment can harm welfare when there is reduced capacity to care for relinquished dogs in
the new temporary or permanent home. Inability to provide for dogs can lead to the morally
challenging decision as to whether euthanasia is chosen, either at the level of the owner who
no longer feels able to care for the dog or those providing management of unowned dog popu-
lations and local authority or non-governmental sheltering organisations [11]. In addition to
the impact on canine welfare, the ethically complex decision to euthanise a dog for the lack of
appropriate care options, however humane the physical act, can cause moral injury to all peo-
ple involved [12]. Euthanasia is also an essential welfare tool, enabling the relief of suffering,
and may be withheld to prolong a human's relationship with their dog. This is especially con-
cerning given the strong owner-dog relationship to dogs with extreme body type [13] that may
necessitate euthanasia. Thus, an increased understanding of different elements of the owner-
dog relationship, and risk factors for weakening of these relationship elements can help to
identify owner-dog dyads at higher risk of poor welfare outcomes.

   To date, research has identified a range of dog- and owner-related factors associated with
the strength of the owner-dog relationship. Owner demographics previously reported as asso-
ciated with a stronger owner-dog relationship include being a younger adult; female; unmar-
ried; having no children; living alone and being the primary care provider for the dog [14–16],
suggesting that time spent together and less division of owner attention is associated with a
stronger relationship [16]. Owners who acquire a dog already confident of their abilities as a
dog owner, such as prior ownership or professional experience, may subsequently have a
stronger relationship with their dog [17].

   Dog characteristics associated with a stronger owner-dog relationship include smaller size
[18], purebred breed status (compared to crossbreeds) [15], and brachycephalic conformation
(with strong owner-dog relationships identified in some extreme brachycephalic breeds: the
Pug, French Bulldog, and English Bulldog) [13]. In contrast, weaker owner-dog relationships
were observed when the only purpose for the dog was companionship compared to working,
sporting or show purposes [19]. The presence of dog behaviours normally considered a prob-
lem by the owner has been associated with a weaker owner-dog relationship [18, 20]. Reward-
based, rather than aversive, training techniques to address or prevent problem behaviours, can
be more effective [21] and can be associated with a stronger owner-dog relationship [22, 23].

In the UK, demand for puppies increased from March 2020 when the government response to the COVID-19 pandemic mandated many people to stay at home during periods of 'lockdown' [24–26]. Compared to owners of pre-pandemic puppies, owners of these 'Pandemic Puppies' purchased during the 2020 phase of the pandemic were reported to be less likely to purchase their first-choice breed of dog, had a shorter interval between deciding to acquire a puppy and bringing them home, and pay more. Pandemic Puppy purchasers were also less likely to follow recommended practice before and during their puppy purchase, with fewer purchasers collecting their puppy in the supplier's home, or seeing the puppy interacting with the puppy's mother [27]. They were also more likely to have risked negative long-term effects on behaviour and health by acquiring their puppy from a poor-welfare source, that is, a breeder or third party supplier who may be selling puppies without due provision for their and their parents' physical, behavioural, and developmental needs and concealing maternal heritable and infectious ill health [28, 29]. Furthermore, essential early-life puppy experiences such as socialisation with humans from outside of their household were thwarted by lockdown restrictions [30], with high levels of problem behaviours reported in this cohort as they reached young adulthood [31]. As this cohort of Pandemic Puppies have matured, there are concerns that the aforementioned behavioural problems, combined with the increased cost of living [32] and return to working away from home compared to when they were purchased in 2020 [33] may increase the burden of pet care for these owners. This raises concerns about the durability of owners' relationships with their dogs, and the subsequent impact on dog welfare and relinquishment [34].

With this background, the current study aimed to identify risk factors associated with a weaker owner-dog relationship in UK dogs acquired during 2020, using the well-established Monash Dog Owner Relationship Scale (MDORS) to quantify this outcome [35]. Greater understanding of these risk factors can be applied to develop better processes and tools to support owners experiencing poor relationships with their dog and to advise future puppy buyers on avoiding risk factors that might endanger a successful relationship with their dog.

## Methods

### Data collection

Survey data were collected via two surveys as part of the Pandemic Puppies project [27, 30]. The project recruited dog owners who were at least 18 years old, resident in the UK, and had purchased and brought home a puppy of any breed or crossbreed under 16-weeks of age during 23 March 2020–31 December 2020 ('Pandemic Puppies'), and a comparator population acquired during the same date-period in 2019 (pre-'Pandemic Puppies'). The first survey, hosted via SurveyMonkey, was open between 10 November and 31 December 2020 and collected data on owner and dog demographics, pre-purchase and purchase motivations and practices, and early-life management and experiences of puppies. A longitudinal study aimed at respondents in the aforementioned Pandemic Puppy cohort (2020 purchases) was launched in 2022, following these puppies as they aged, with data collection timepoints in young adulthood at 21-, 24-, 27- and 36-months of age. These surveys collected data on current dog management, health, behaviour, and the owner-dog relationship. This current study explores data reported from the first survey in 2020 and the 21-month timepoint that was collected during 2022. The outcome of interest for this study, the owner-dog relationship, was measured via the MDORS [35], as described below. The survey was hosted using a version of the Vanderbilt University's Research Electronic Data Capture platform (REDCap) provided by the Royal Veterinary College, a part of the REDCap Consortium [36, 37].

Ethical approval for the Pandemic Puppies study was granted by the Social Science Research Ethical Review Board at the Royal Veterinary College (SR2020-0259), with written informed consent supplied by all respondents. The full 2020 questionnaire has been published previously [27, 30]. Questions relevant to the current study from the 21-month survey can be found in S1 Appendix.

## Survey content and data coding

**Owner-dog relationship.** MDORS is a psychometric scale for the measurement of the owner-dog relationship consisting of 28 items grouped into three subscales: Dog Owner Interaction (hereafter referred to as Interaction), Perceived Emotional Closeness (hereafter referred to as Closeness), and Perceived Costs. As described previously [35], respondents were offered fixed-choice response options for each MDORS item which were allocated a value from 1 to 5 where 1 represented strongly disagree or least frequent and 5 represented strongly agree or most frequent. Item values were summed to produce a score for each subscale. Possible scores for Interaction subscale ranged from 9 to 45, where higher scores represented more shared activities between the owner and their dog. Possible scores for Closeness subscale ranged from 10 to 50, where higher scores represented an owner feeling closer to their dog. Possible scores for Perceived Costs subscale ranged from 9 to 45, where higher scores represented an owner feeling more burdened by dog ownership. Where any of the individual 28 MDORS questions were not answered, all of that respondent's data were excluded from analysis in the current study.

**Risk factor variables.** Risk factors for a weaker owner-dog relationship collected from the surveys and evaluated during risk-factor modelling are listed in Table 1. Risk factors included were:

i. those considered risk factors for owner-dog relationship based on existing literature or plausible association;

ii. risk factors associated with being a 'Pandemic Puppy' (purchase-related and early-life factors found to be significantly different in prevalence in the 2020 Pandemic Puppies cohort compared to pre-'Pandemic Puppies') [27, 30];

iii. UK legal requirements and proposed best practices and for puppy acquisition; and

iv. risk-factors plausibly influenced by economic and public health events at the time of the surveys [25, 32, 33].

## Pre-purchase and purchase motivations and practices

Respondents were asked in the 2020 survey about their motivations, and pre-purchase and purchase behaviours for when they bought their dog. The main reasons (more than one permitted) for dog purchase were recorded with a yes or no response to each of several motivations for purchase. Purchase motivation options indicating companionship for self, children, or other adults were used to generate the dummy variable for when the dog was acquired for companionship of others in the household, but not the respondent. Respondents were asked if their household carried out research before they purchased their puppy, with the response options: yes, no, or 'no but I am already an experienced dog owner'.

## Dog demographics

Owner reported data on their dog's breed were used to generate derived variables on a range of demographic factors. Applying prior work on cleaning and mapping breed terms from

**Table 1. Potential risk-factors for weaker owner-dog relationship.**

| Potential risk factor | Primary reason to consider* |
|---|---|
| **Respondent demographics** | |
| Owner age (years)[a] [18, 19, 38] | Identified |
| Owner gender[a] [18] | Identified |
| Region within UK[a] [25] | COVID-19 |
| Owner employed in animal care sector[a] | Pandemic Puppy |
| Owner does not live with another adult[b] [16, 18, 19] (unmarried: [15]) | Identified |
| Children in the home[a*] [15, 16, 19] | Identified |
| **Household characteristics** | |
| Number of other dogs in the home[a] [16, 19] | Identified |
| A dog was present in the owner's home whilst they were a child[a] [19] | Identified |
| Owner has previously owned or co-owned a dog as an adult[a*] [19] | Identified |
| Owner's work location[b] | COVID-19 |
| Other adult in household works away from home[b] | COVID-19 |
| COVID-19 affected finances[b] [39] | Identified |
| Anticipated changes to household circumstances in next three months[b] [39] | Identified |
| **Acquisition motivation and preparation** | |
| Owner acquired dog motivated by 'companionship for myself'[a] [19] | Identified |
| Owner acquired dog motivated by 'companionship for other people in household, not including respondent'[a] [19] | Identified |
| Owner acquired dog as working dog for specific role[a] [19, 40] | Identified |
| COVID-19 pandemic influenced respondent's decision to acquire a dog[a] | COVID-19 |
| Owner performed research on dog ownership prior to acquisition[a*] [38] | Identified |
| Length of time from decision to acquire to bringing dog home[a] | Pandemic Puppy |
| **Acquisition characteristics** | |
| Owner acquired first choice breed/crossbreed[a] | Pandemic Puppy |
| Purchase price category (GB£)[a] | Pandemic Puppy |
| Collected inside breeder's home[a] | Pandemic Puppy |
| Puppy seen with mother at collection[a] | Pandemic Puppy |
| Microchip provided by breeder[a] | Law |
| Passport provided by breeder (for dogs over 15-weeks only)[a] | Pandemic Puppy |
| **Dog characteristics** | |
| Dog's sex | Plausible |
| Dog's neuter status | Plausible |
| Dog's typical adult bodyweight (kg)[a] [18] | Identified |
| Birth month of dog (year 2020)[a] [25] | COVID-19 |
| Dog's breed designation (crossbred, purebred, or designer crossbred)[a*] [15, 16] | Identified |
| The Kennel Club (UK) breed group of dog, including 'not recognised'[a] [19, 41] | Identified |
| Dog is a Pug, French Bulldog or English Bulldog[a] [13] | Identified |
| **Dog management and health** | |
| Owner primary carer of dog in 2020[a*] [16] | Identified |
| Change in who is involved in dog's care from 2020 to 21-months old[b] | COVID-19 |
| Puppy left alone > 4 hours in 2020[a] [25] | COVID-19 |
| Dog left alone > 4 hours at 21-months of age[b] | COVID-19 |
| Owner takes dog if work outside the home[b] [42] | Identified |
| Respondent attended puppy classes with their dog whilst they were < 16-weeks old[a] | COVID-19 |
| Owner attended adult (> 16-weeks old) dog training classes with their dog[b] | COVID-19 |
| Training methods used by owner[b] [23] | Identified |

*(Continued)*

**Table 1.** (Continued)

| Potential risk factor | Primary reason to consider* |
|---|---|
| Dog has ongoing health problem requiring veterinary attention[b] [13] | Identified |
| **Dog behaviour** [18, 19, 43] | Identified |
| Control problem[b] | |
| Attention seeking[b] | |
| Aggression[b] | |
| Fear/avoidance[b] | |
| Reaction to familiar people[b] | |
| Reaction to other dogs[b] | |
| Abnormal repetitive behaviours[b] | |
| Separation related behaviours[b] | |

Potential risk factors for a weaker dog owner bond (as quantified by the Monash Dog-Owner Relationship Scale scores) in a population of Pandemic Puppies aged 21-months and bought as puppies under 16-weeks of age in the UK between July to December 2020.

[a]Data from Pandemic Puppies 2020 survey

[b]Data from survey when same dogs reached 21-months-old

* Reasons for inclusion:

Identified: Risk factors identified in existing literature

Law: required by law in the United Kingdom

Pandemic Puppy: Risk factor significantly different in prevalence in the 2020 Pandemic Puppies cohort compared to pre-'Pandemic Puppies' cohort [27, 30]

COVID-19: Risk factor plausibly influenced by economic and public health events at the time of the surveys

VetCompass [44], breed terms were linked to derived variables for: 1) The Kennel Club (UK) breed groups; 2) typical adult bodyweight (kg); 3) identification as either Pug, French Bulldog, or English Bulldog due to prior interest based upon the work of Packer et al [13]; and 4) breed designation (purebred, crossbred, or designer crossbred—the latter defined as purpose-bred crosses between defined purebred progenitor breeds [45]).

## Dog management, training, and health

Respondents reported in the 2020 survey whether they were the primary carer for their dog or if care was shared within or outside the household. In the 21-month survey, respondents were asked if there had been a change in who was involved in their dog's care, including dog walkers or doggy day-care, and whether the owners left their dogs for more than four hours was reported in both surveys. In the 21-month survey, respondents indicated their current work location (in relation to home); whether any other adult in the household works away from home; whether they took their dog to their place of work; whether COVID-19 affected the household finances; and whether the respondent anticipated that changes in household circumstances would make it easier or harder to own a dog.

Attendance was reported for puppy classes at less than 16-weeks-old, and adult training classes to 21-months-old (in person, online, or no). Respondents reported training methods or aids that they used on their dog up to 21-months of age. Based on the principles of operant conditioning, each method or aid was classified was as reward-based (positive reinforcement, negative punishment) or aversive (negative reinforcement, positive punishment). Respondents were allocated a training style based on the classification of methods or aids they used as reward-only, aversive-only, rewards with one aversive method, or rewards with more than one aversive method [31].

In the 21-month survey, respondents who were registered at a veterinary practice, and had been seen by a veterinary professional since the 2020 survey, were asked if their dog had any ongoing health problems that required ongoing veterinary care, excluding routine preventative care. Respondents were asked if their dog had been neutered in 2020 and 2022, with their most recent neuter status used in this analysis.

## Dog behaviour

Problem behaviours considered as risk factors were owner-reported problem behaviour, separation related behaviour, and abnormal repetitive behaviour. Questions regarding owner-reported problem behaviours and separation-related behaviours (SRBs) used in surveys of this Pandemic Puppy cohort have been previously summarised and their risk factors analysed [31]. These variables were further analysed in the current study as risk factors for MDORS subscale scores. Briefly, when dogs were 21-months old, respondents selected behaviours from a fixed-choice list that they considered problematic in their dog, with a free-text option if their specific problem behaviour was not listed. These owner-reported problem behaviours were categorised by likely emotional-contextual origin as: attention seeking, aggression, fear and/or avoidance, reaction to familiar people, reaction to other dogs, and problems related to owner control [31]. The number of specific problem behaviours within each emotional-contextual origin category was reported for each dog.

Respondents also selected behaviours displayed by their dog from a list of nine SRBs together with which of two context options the potential SRB was displayed in: while the dog was with people who were at home relaxing; or while the dog was left at home alone. SRB cases were defined as dogs with at least one reported SRB displayed when home alone [31]. Non-SRB cases were defined as dogs that displayed either none of the nine potential SRBs, or only displayed potential SRBs while the dog was with people who were relaxing within the home.

In addition, respondents reported if any of five abnormal repetitive behaviours (ARBs) were displayed by their dog: circling, shadow-chasing, fly-snapping, oral behaviours (e.g., prolonged self-directed licking or licking/sucking of other objects), or pica (see S1 Appendix for complete questionnaire wording). The number of individual ARBs displayed per dog was summarised as a predictor variable.

## Data coding

All risk factors were treated as categorical variables. For questions that offered both fixed-choice and free-text responses, data from free-text responses that matched existing fixed-choice responses were manually backcoded, as described previously [27]. If no response was provided for a risk-factor question, "No answer" was included as a category for that factor. Where appropriate, categories containing fewer than ten observations were combined with adjacent categories if both categories were ordinal numerical data; or if both categories indicated uncertainty. Where merging was inappropriate, categories with few observations were retained rather than discarded to avoid reduction of sample size.

Questions that offered fixed-choice responses describing both actions (e.g., whether the owner took their dog to work) and motivations (e.g., why they did not take their dog to work) were collapsed to actions only. Information from a prior question asking whether people could not remember the price paid for their puppy, or would prefer not to say, were added as categories to the price variable.

## Analysis

All three MDORS subscale score distributions were not normally distributed on inspection of histograms and subsequently reported as median, interquartile range and range. A Cronbach's

alpha statistic was calculated for each MDORS subscale to determine reliability (internal consistency) in this cohort. A minimum Cronbach's alpha of 0.7 for each subscale was considered acceptable reliability [35, 46] to proceed to risk factor analyses.

Sample size estimation using OpenEpi [47] indicated that scores from at least $n = 148$ respondents were required to detect a MDORS subscale score difference of 3.00 between two groups (standard deviation 6.00) where the size of one group was three times that of the other.

## Multivariable modelling

Risk factor analysis used separate backwards stepwise multivariable linear mixed modelling for each of the three MDORS subscale outcome variables (Interaction, Closeness and Perceived Costs) provided these showed acceptable reliability from the Cronbach's alpha testing. Respondent age and gender were retained in the final models as *a priori* confounders, and UK region and dog birth-month were included as random effects to account for different external events affecting dog ownership in the latter half of 2020. Risk factors with liberal univariable association ($p < 0.2$) with the separate MDORS subscale outcomes using an F-test were carried forward to the maximum multivariable models. The distribution of standard error of MDORS scores was assumed to be normal, due to central limit theorem [48].

Pearson's coefficient of $> 0.7$ and variance inflation factor $> 10$ were used to identify possible collinearity between risk factors. Where collinearity was identified, the variable considered the most useful explanatory variable was chosen for inclusion in the modelling [46]. At each step during model building, the risk factor with the highest partial F-test $p$-value of at least 0.05 was eliminated [46]. Model comparisons were conducted with Akaike Information Criterion [49, 50]. If elimination of a risk factor resulted in greater than 25% change to the coefficients of at least one remaining variable, then the risk factor was retained in the model [19]. To show the standardised mean difference of an effect, Cohens $d_s$ effect size for the sample was calculated for categories included in the model with a z-test $p$-value $< 0.05$ [51]. Microsoft Excel was used for initial data cleaning and Stata 18SE was used for statistical analysis and charts.

## Results

### Study sample

A total of $n = 1007$ respondents responded to both the original (puppyhood) and first follow up survey (21-months) of the Pandemic Puppies project. Of these, $n = 985$ (97.82%) respondents still owned their dog, $n = 13$ dogs (1.29%) had been rehomed or sold, and $n = 9$ (0.89%) dogs had died or been euthanised. Of respondents who still owned their dog, $n = 978$ (99.28%) elected to proceed with the full 21-month survey. Complete responses for all 28 Monash Dog-Owner Relationship Scale (MDORS) items were provided by $n = 794$ (81.19%) respondents. Incomplete MDORS responses by $n = 184$ (18.81%) consisted of $n = 129$ (13.19%) respondents who answered no items, and $n = 87$ (8.90%) respondents who omitted at least one item. Respondents with incomplete MDORS responses were excluded from analysis (see Fig 1).

### Descriptive statistics

**Demographics.**   In the subset of 794 respondents with complete MDORS responses the majority ($n = 719$, 90.55%) of respondents identified as female and all age range categories and UK regions were represented. Most dogs were categorised as either purebred ($n = 551$, 69.40%) or designer crossbred ($n = 211$, 26.57%), with a minority being crossbreeds ($n = 32$, 4.03%). Very few dogs were one of three brachycephalic breeds previously associated with a high MDORS scores: Pug, French Bulldog or English Bulldog (total $n = 13$, 1.64%).

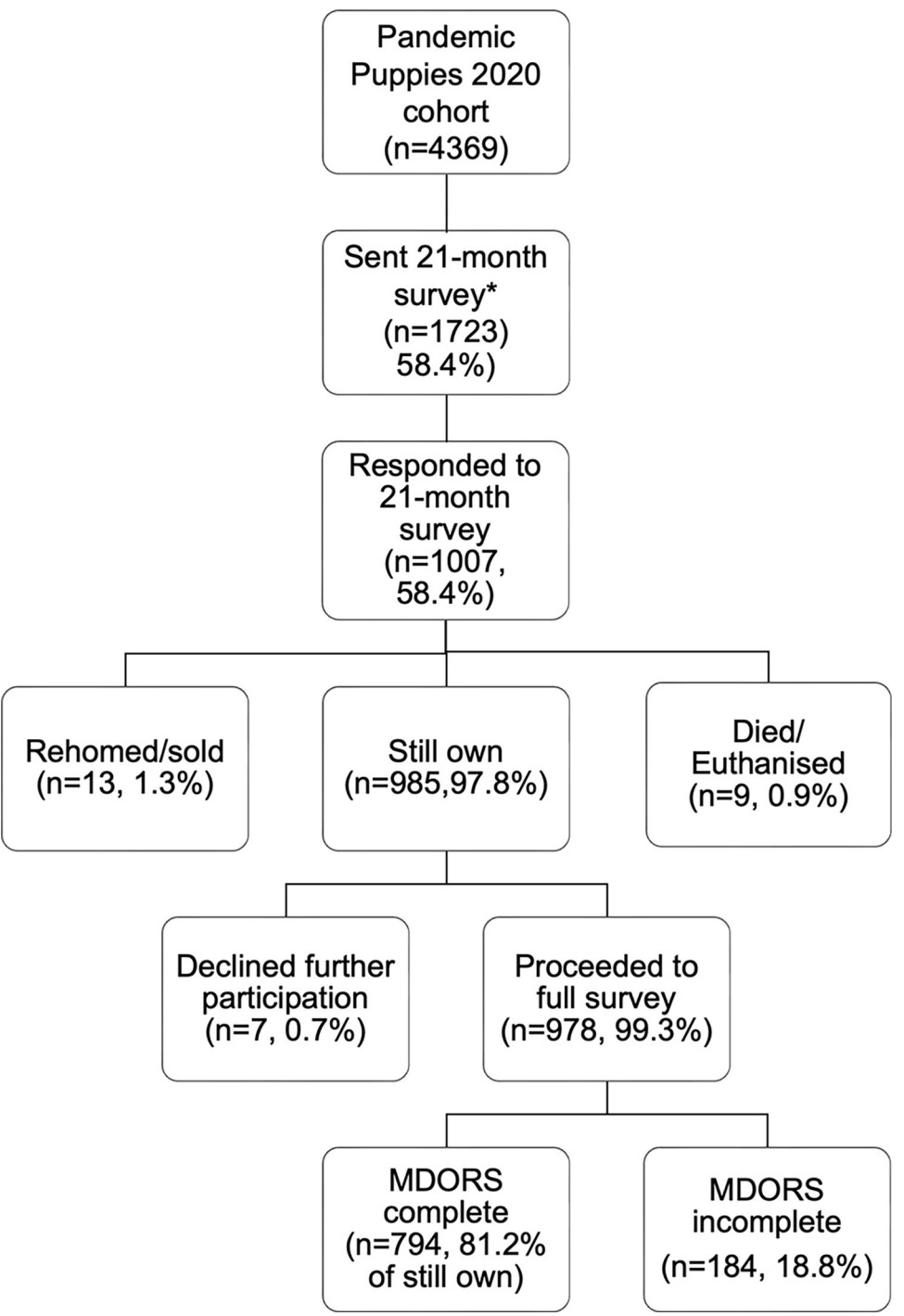

**Fig 1. Sample size and response attrition for investigation of risk-factors for weaker owner-dog relationship measured by the Monash Dog-Owner Relationship Scale (MDORS) as reported by owners of Pandemic Puppies aged 21-months, bought as puppies under 16-weeks of age in the UK between 1 July to 31 December 2020.**

**Lifestyle and behaviour.** Nearly half of respondents ($n = 359$, 45.21%) did not take their 21-month-old dog with them to work outside the home, and a further 43.83% ($n = 348$) of owners did not leave the home to work. Over half of owners used a mixed training style which

**Table 2. Summary Monash Dog-Owner Relationship Scale (MDORS) scores reported by owners of Pandemic Puppies aged 21-months, bought as puppies under 16-weeks of age in the UK between July to December 2020 ($n$ = 794).**

| Subscale (possible range) | Range | Median | Interquartile range | Cronbach's alpha |
|---|---|---|---|---|
| Dog owner interaction (9–45) | 18–45 | 38 | 36–45 | 0.523 |
| Perceived emotional closeness (10–50) | 14–50 | 40 | 35–46 | 0.886 |
| Perceived costs (9–45) | 9–38 | 16 | 12–20 | 0.871 |

included two or more aversive methods ($n$ = 470, 59.19%). Most respondents reported at least one problem behaviour related to control ($n$ = 649, 81.74). A full list of risk factors, the proportion of responses in each category, and univariable linear regression results for each separate outcome, is in S2 Appendix.

**MDORS scores.** Median MDORS scores were towards the higher end of the range for Interaction and Closeness subscales, and the lower end of the Perceived Costs subscale (Table 2).

Frequencies of responses to each MDORS item and histograms of subscale scores are in S3 Appendix.

Cronbach's alpha exceeded 0.7 for the Closeness and Perceived Costs subscales, indicating high internal reliability of these scores. The Interaction subscale did not meet the threshold for reliability in this analysis (Cronbach's alpha: 0.523) and was thus not analysed further in this study.

## Multivariable model building

Four of the 49 risk factors explored did not have a liberal univariable association ($p < 0.2$) with either of the Closeness and Perceived Costs subscale outcomes and were therefore not considered in multivariable modelling: attendance at training classes (puppy and adult dog); owner leaving their puppy alone for more than four hours; or microchip having been provided by the breeder. The remaining variables met the liberal association criterion with either or both Closeness and Perceived Costs scores. All behaviour variables had a highly statistically significant univariable association ($p < 0.003$) with the Perceived Costs subscale.

Pearson pairwise correlations coefficients were $< |0.5|$ in both models, and Variance Inflation Factors (VIF) were $< 10$ for Closeness. In the case of Perceived Costs, VIF were $> 10$ for The Kennel Club (UK) breed group and breed type designation, therefore breed group was excluded from the modelling. Final linear mixed models for both Closeness and Perceived Costs showed normality of residuals and homoscedasticity by inspection of plots (see S4 Appendix).

Adjusted R-squared for a fixed effects model of Perceived Costs scores was 0.21. Random effects for UK Region and dog birth-month improved the fit of the Costs model (likelihood ratio test $p$ = 0.040) compared to a fixed-effects multivariable linear regression model.

Adjusted R-squared for a fixed effects Closeness model was 0.19. The random effects model was reported, however no improvement was detected from adding UK Region and dog birth-month as random effects to the Closeness model (likelihood ratio test $p$ = 0.996).

## Final multivariable model for Perceived Costs

Problem behaviours related to lack of control, fear, aggression, and separation represented many of the risk factors significantly associated with increased Perceived Costs score, after adjusting for UK Region and the dog birth-month, when more than one different problem behaviour was reported within each motivation-context of control, fear, and aggression (Fig 2).

Two risk factors related to motivation for purchase were associated with increased Perceived Costs scores: dogs acquired as a companion for others in the household rather than the respondent; and the COVID-19 pandemic explicitly influencing the puppy purchase (Fig 2).

Factors associated with reduced Perceived Costs score related to the provision of care by owners for their dog, and their prior experience before acquiring this dog (Fig 2). Respondents who stated they were experienced dog owners as the reason for not doing research prior to purchasing their puppy had a lower Perceived Costs score compared to those who reported that they did prior research before purchasing their puppy (Fig 2). Perceived Costs scores were lower among respondents who reported the primary caregiver for their dog had changed since

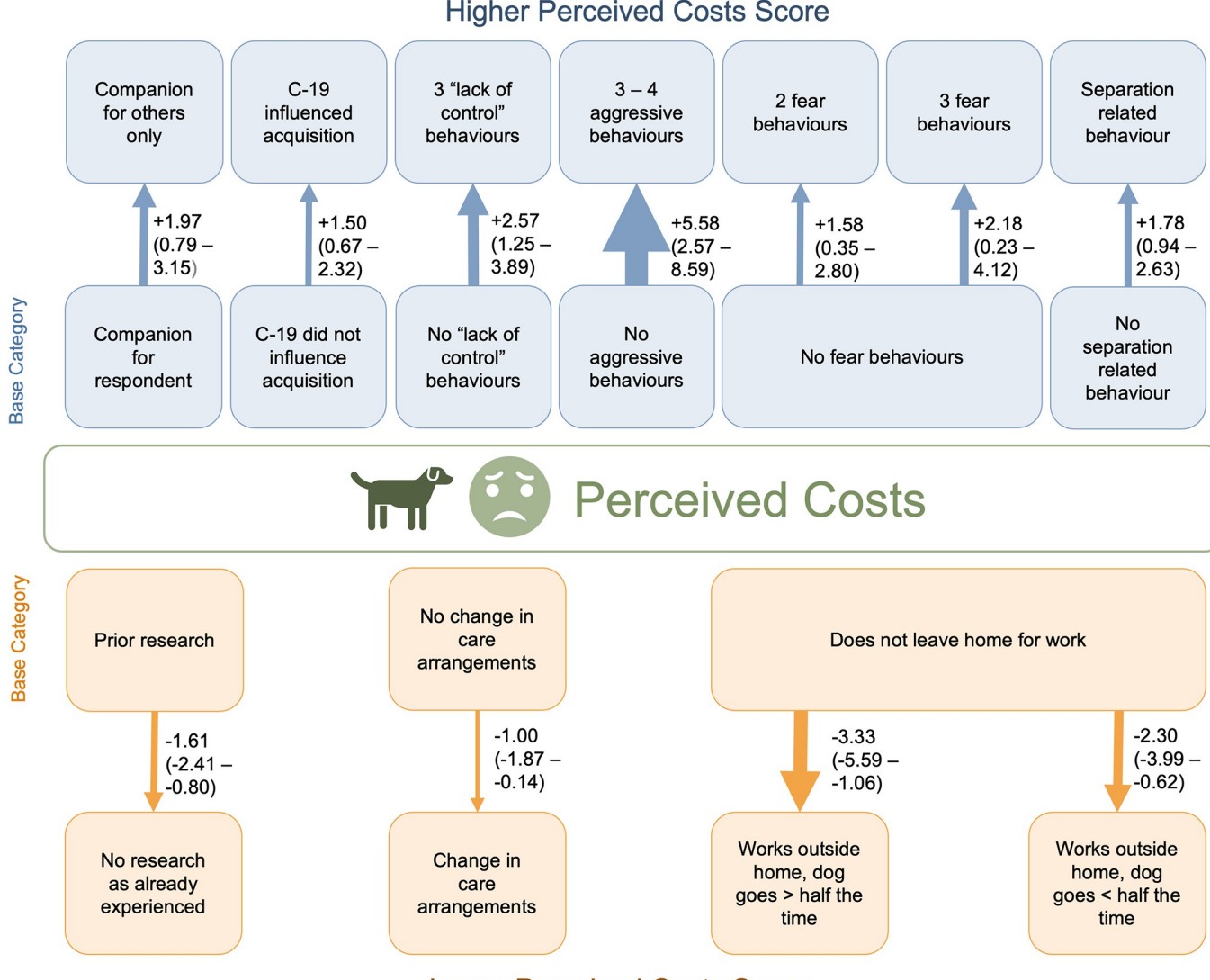

**Fig 2. Mixed multivariable linear regression model for risk factors influencing Perceived Costs to owners of 21-month-old Pandemic Puppies bought as puppies under 16-weeks of age in the UK between July to December 2020, as measured by the Perceived Costs subscale of the Monash Dog-Owner Relationship Scale (*n* = 794).** Blue/up arrows indicate risk factors that increase Perceived Costs reflecting a weaker owner-dog relationship, orange/down arrows display decreased Perceived Costs reflecting a stronger owner-dog relationship. Arrow width proportional to effect size. Regression coefficient (95% confidence interval) represented to the right of the arrow for each risk factor. Risk factors with an effect size with a confidence interval that includes 0 are not included. Adjusted for training method and anticipated changes to household circumstances. Random effects: UK Region and dog birth month.

the first survey compared to those reporting no change, and for respondents who took their dog with them to work (if they worked outside the home), compared to those that did not leave the home to work (Fig 2).

Training methods and anticipated changes to household circumstances in the next three months were retained in the model with a significant contribution based on the partial F-test, but no categories showed a significant effect size.

Largest effect sizes (those categories with the highest standardised magnitude of change to the outcome, Perceived Costs, compared to their base category) were observed as an increase in Perceived Costs when aggressive behaviours were reported in 3 or 4 different contexts, such as towards their owner, strangers, other dogs (Cohens $d_s$ +0.95, 95% confidence interval (CI): 0.40–1.51); and a decrease in Perceived Costs when the respondent took their dog to work outside of the home at least half of the time, compared to not leaving the home for work (Cohens $d_s$ -0.57, 95% CI: -1.00 to -0.14) (Fig 2).

## Final multivariable model for Perceived Emotional Closeness

Risk factors associated with changes to Closeness scores were largely related to dog and owner demographics, dog purchase, and management of the dog, after adjusting for UK Region and dog birth-month. Aggression was the only statistically significant behavioural risk factor in the Closeness model (Fig 3).

Many demographic risk factors were associated with Closeness scores. Closeness scores were lower for men compared to women; for respondents aged 35–44 or 55–64 years old compared to those aged 25–34 years old; for dogs which weighed 10–30 kg compared to those 0–10 kg; for respondents with children in the home compared to none; and respondents working a mixture of at home and outside the home compared to working solely from home (Fig 3). Living alone compared to living with another adult was associated with higher Closeness scores, as was another other adult working outside the home compared to working at home, and change (for better or worse) in household finances compared to those prior to the COVID-19 pandemic (Fig 3).

Closeness scores were higher when the dog had been acquired as a companion for the respondent; acquired for a specific working role; acquired within six months of deciding to own a dog compared to taking longer to decide; and if the respondent had previously owned a dog as an adult (Fig 3). Closeness scores were lower when primary care for the dog was shared with another individual living separately from the respondent's household, and higher if respondents never took their dog to work with them outside the home compared to not leaving the home for work (Fig 3). Using only one aversive-based method compared to two or more aversive-based training methods was associated with increased Closeness scores (Fig 3).

Risk factors associated with lower Closeness scores with notable effect sizes were the primary caring being shared with another individual living separately from the respondents household compared to the respondent being the primary caregiver for the dog (Cohens $d_s$ -0.77, 95% CI:-1.28—-0.25) and if 3 or 4 different types of aggressive behaviour were reported (Cohens $d_s$ -0.77, 95%CI -1.32 –-0.22) (Fig 3). Duration of acquisition decisions was also significantly associated with Closeness scores, with more rapid decisions associated with higher scores; taking less than a week between the decision to acquire a puppy and bringing them home was associated with a higher Closeness score with a large effect compared to those who took six months or longer (Cohens $d_s$ +0.62, 95% CI 0.10–1.14) (Fig 3). Although smaller effect sizes, the same direction of relationship was seen between those who took between 1 week– 1 month (Cohens $d_s$ +0.48, 95% CI 0.20–0.77) or 1–6 months (Cohens $d_s$ +0.16, 95% CI 0.01–0.30) between decision to acquire to bringing their puppy home, compared to taking six months or longer.

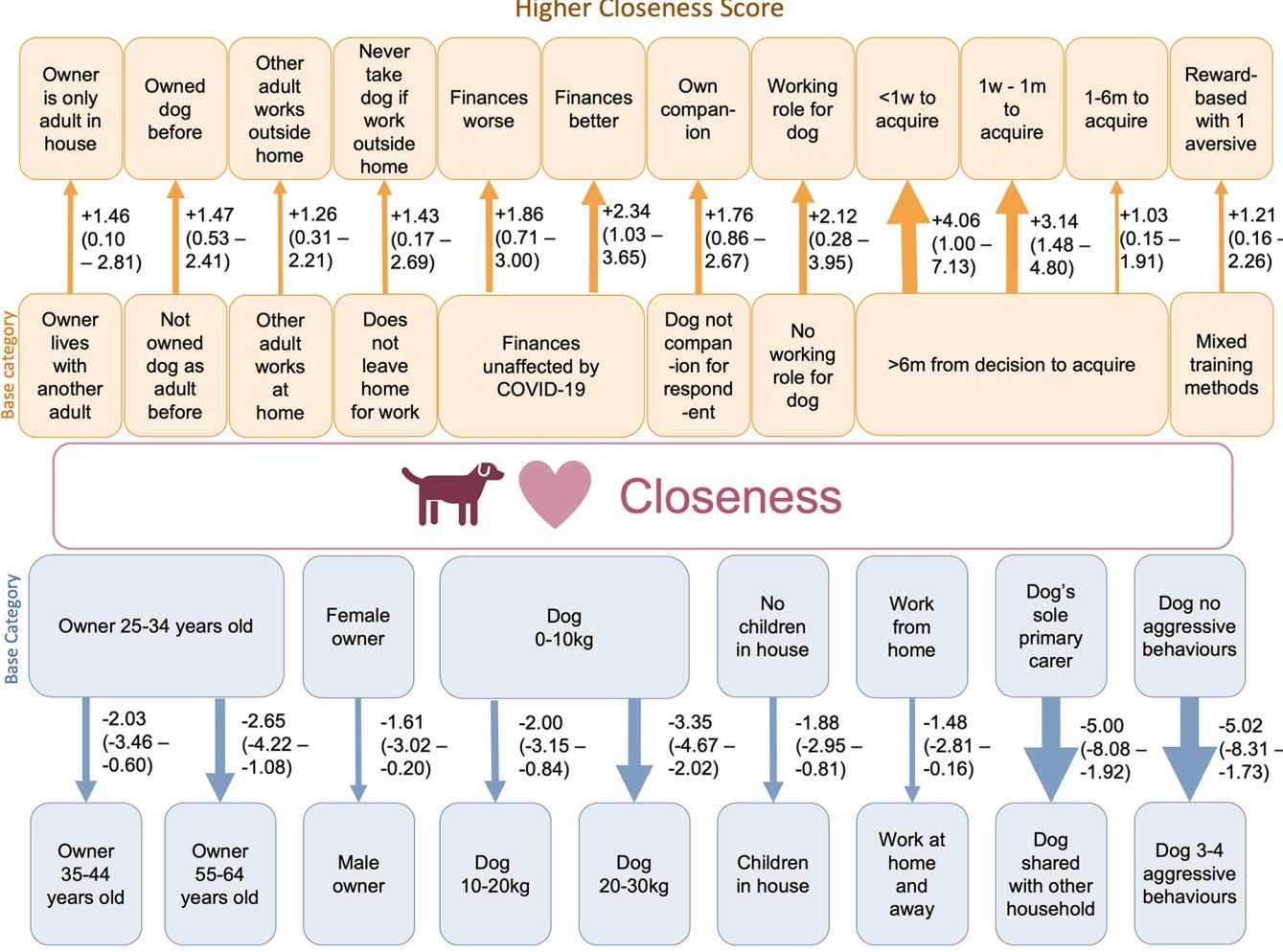

**Fig 3. Mixed multivariable linear regression model for risk factors influencing Perceived Emotional Closeness between owners of 21-month-old Pandemic Puppies bought as puppies under 16-weeks of age in the UK between July to December 2020, as measured by the Perceived Emotional Closeness subscale of the Monash Dog-Owner Relationship Scale (*n* = 794).** Orange/up arrows indicate risk factors that increase Closeness reflecting a stronger owner-dog relationship, blue/down arrows display decreased Closeness reflecting a weaker owner-dog relationship. Arrow width proportional to effect size. Regression coefficient (95% confidence interval) represented to the right of the arrow for each risk factor. Risk factors with an effect size with a confidence interval that includes 0 are not included. Random effects: UK Region and dog birth month.

## Discussion

Despite earlier fears of high levels of relinquishment for puppies acquired during the early part of the pandemic [24, 34], few dogs in this study had been relinquished by 21-months of age. However, measurement of the owner-dog relationship in the current study has still identified risk factors, both pandemic and non-pandemic related, that can negatively impact owner's emotional relationship with their dog and increase the burdens of dog ownership. If not addressed, these factors could impair the welfare of dogs, reduce the wellbeing of owners, and lead to relinquishment and euthanasia of dogs in the future.

This current study highlights the association of problem canine behaviours with increased burdens of dog ownership; reveals complexity between owner's relationships with their dogs and work commitments; and confirms that demographic risk factors represented many of the

factors influencing the Closeness score—including owner age [18, 19, 38], gender [18] and owners having no children in the house [15, 16, 19].

## Dog behaviour and Perceived Costs

Owner-reported problem behaviours were identified as the primary drivers of increased Perceived Costs of dog ownership, a negative element of the owner-dog relationship reflecting the 'burden' of caregiving. Problem behaviour has previously been reported as a risk factor for a reduced owner-dog relationship [18, 43], relinquishment [52] and euthanasia [53]. As aforementioned, although in the current study only 22 owners who responded at 21-months had either relinquished their dog or had their dog euthanised, four of the 13 relinquishments and three of the nine euthanasias were due to behavioural issues [31]. Future behavioural relinquishments and euthanasia may occur in pandemic puppies if problem behaviour is exacerbated or cannot be resolved, given that behaviour problems are most commonly presented in dogs aged 2.5–3.7 years of age [54], and there is a proportionally high risk of euthanasia for undesirable behaviour at this age [53]. Aggression towards people and other animals have been identified as the main drivers of behavioural euthanasia in previous international studies [55–57], and thus particular focus on this population of dogs and their owners is needed by researchers, and also from organisations offering practical support. Longitudinal study of this population may provide insight as to whether resolution of problem behaviour can decrease the Perceived Costs of dog ownership, and thus reduce relinquishment and euthanasia on behavioural grounds, or whether persistent or worsening behavioural problems are associated with these negative outcomes.

The current study demonstrates the differential effects of problem behaviour on different facets of the owner-dog relationship: problem behaviour is shown as a major contributor to the Perceived Cost of pet ownership but has a lesser impact on the owner's feelings of emotional closeness towards that dog, with the notable exception of aggressive behaviours in multiple contexts. Aggression has the potential to reduce many of the social activities that people find pleasurable with their dog, both inside the home and in the community, thus reducing the opportunity for shared time and development of the emotional relationship [16]. It has been suggested that lower attachment from their owner may lead to increased dog aggression [58], or alternatively the fear of harm to self, other people, and other dogs may have made owners feel insecure in their relationship with their dog and therefore reduced the closeness of the owner to their aggressive dog. A Danish study measuring MDORS [19], also found owner perceived problems with fear were associated with increased Perceived Costs but not Closeness. Aggression or being home alone were not shown to be associated with either outcome, perhaps reflecting the different cultural expectations and management of dogs.

Other studies of showing the relationship between problem behaviour and the owner-dog relationship have examined the interaction between the owner and the dog [18], or attachment [10, 43], and thus have not been able to identify the contrasting effect of behaviour on different aspects of the relationship. The low frequency of relinquishment in this study could be further explained by problem behaviour not being associated with decreased Closeness (and thus the close emotional relationship with the dogs buffers the 'burden' of owning them resulting in less relinquishment [10]). However, it has been shown that relinquishing owners do not necessarily have lower emotional attachment to their dogs [39] so more information on the changes to the owner-dog relationship over time in relation to risk factors and relinquishment is warranted.

That owners are emotionally close to their dogs despite problem behaviour does not mean the burden of problem behaviour is not of wider concern. Some behaviours that were reported in the current study, such as vocalisation or toileting indoors, can indicate a poor emotional

state in the dog displaying them [59] rather than merely normal canine behaviour that is unwanted by the owner. Problem behaviour in dogs can negatively affect owners' mental health [7], potentially through the mechanism of disenfranchised guilt [9], and thus even though owners may still enjoy a close relationship with their dog, it may have an impact on the owners' mental wellbeing. Given that problem behaviours were found to be widespread in this cohort (96.7%; reported in a sister paper exploring behaviour in the Pandemic Puppy cohort at 21-months [31]), evidence-based interventions aimed at reducing problem behaviour are likely to be vital in improving and protecting dog welfare as well as reducing associated burdens of dog ownership, and thereby restoring the owner-dog relationship.

## Training

Only the use of mainly reward-based training (with one aversive) compared to those who used multiple aversive training methods was associated with a stronger relationship (higher Closeness, lower Perceived Costs), and, of these, only the association with Closeness had a 95% confidence of a positive effect size. No association was detected for purely reward-based training, and no respondents used only aversive training methods. The most common aversive training method used by respondents was physically moving their dog (e.g., to move hindquarters to encourage sit, or move off furniture) [31], but additional data on the degree of force and emotional reaction of the dog was not collected. Given that reward-based training has previously been shown to be more effective [60], and associated with less problem behaviour in this sample [31], the inclusion of problem behaviour in the model has reduced statistically the association between relationship and training methods. The observation of a weaker owner-dog relationship with increasing use of aversive techniques in this current study supports current evidence promoting reward-based training [21, 22, 60]. This is an important result given the high prevalence of the use of aversive dog training techniques among owners in this [31] and other study populations [21, 60, 61].

## The COVID-19 pandemic and the owner-dog relationship

Of those risk factors tested that were related to changes in dog acquisitions during the pandemic [27, 30], several were found to be associated with MDORS outcomes. COVID-19 explicitly influencing the owners' decision to acquire a dog was associated with increased Costs: this may reflect these decisions being based on the specific lifestyle of owners during this atypical time, but that once changed, led to a higher burden of ownership. Given that approximately 40% of owners felt that the COVID-19 pandemic influenced their decision to acquire a puppy in 2020 [27], a large number of dog-owners nationally may be affected by this additional burden. It follows that public messaging and behavioural interventions, e.g., from animal charities, should focus on reaching future owners, considering the potentially 11-year average lifespan [62] of UK dogs, and to reflect on potential lifestyle changes in this 11-year period that could impair their ability to care for their dog in the medium-long term. However, efforts to encourage 'responsible' acquisition, where prospective owners are aware of the responsibilities that come with owning a dog and consider their ability to care for a dog for the whole of the dog's lifespan, appear to have limited effectiveness in the UK. For example, the charity Dogs Trust's famous slogan, 'A Dog is for Life, not just for Christmas' was repurposed for the pandemic as 'A Dog is for Life, not just for Lockdown', and yet the Pandemic Puppy phenomenon was widespread [63]. Studies report that owners' awareness of the legal responsibilities that come with dog ownership are poor, including those related to safeguarding health and welfare. For example, Irish dog owners were no more knowledgeable than non-dog owners regarding the responsibilities of dog owners prescribed by law in Ireland [64]. Consequently, greater

understanding is needed of why owners continue to impulsively acquire dogs in the face of such widespread messaging, when the owners' future circumstances are liable to lead to challenges providing care; and ultimately even relinquishment. Restrictions to current acquisition practices (e.g., compulsory courses that new owners must attend, as already implemented in countries such as Spain since 2023 [65] and in the Canton of Zurich since 2022 [66]) could be an effective deterrent for impulsive acquisition. However, the efficacy of these rules and whether there would be political will and resources to implement them in the UK is currently unknown.

COVID-19 affecting household finances, for better or worse, was associated with increased Closeness. Both directions of change may have led to the respondent and their dog spending more time together strengthening the relationship [67], either through more time spent at home (rather than paid work) or ability to participate in shared paid-for activities.

A shorter period between deciding to acquire a puppy and bringing them home was significantly associated with increased Closeness. This could be a reflection of owners acting to prevent the pandemic thwarting a long-held desire to get a puppy, rather than impulsivity, although the pandemic influencing the decision to acquire a dog was a separate risk factor. It is possible the strong relationship between Closeness score and a short interval between decision and acquisition of a puppy may be because both are a measure of the owner's expression of emotion, rather than rapid acquisition genuinely leading to forming a stronger relationship. However, Closeness score has been previously associated with the human personality trait "conscientiousness" [41], which is not consistent with this hypothesis.

### Owner working and dog-care arrangements

Risk factors related to management of the dog (work location, taking dog to work, change in care arrangements) showed varied and superficially contradictory associations with either outcome of Closeness or Perceived Costs. For example, compared to working at home, taking the dog to work outside the home was associated with reduced Perceived Costs (stronger relationship; perhaps due to the increased efficiency of working while caring for their dog at no extra financial cost, e.g., daycare). However, working outside the home address without the dog was associated with increased Closeness (also stronger relationship; perhaps due to the relative novelty of interactions between dog and owner compared to those together all day, or reduced conflict between the demands of paid work and caring for their dog). Also, not sharing primary care was associated with increased Closeness (stronger relationship) and changing who was involved in the care of the dog since 2020 was associated with reduced Perceived Costs (also stronger relationship). The apparent contradiction could be explained by interaction between variables, whereby subpopulations of owners within this sample may have had different motivations for working location and dog management which could not be identified with this sample size. The complex relationship between owners' feelings for their dogs, burdens of pet ownership, and commitment to work warrants further investigation, particularly as work-family conflict has been identified as a stressor in dog owners [9].

### Limitations

The respondents of the Pandemic Puppies surveys were highly skewed towards women, as is common in survey research about pets [7, 8, 13, 35]. The dog's primary caregiver was asked to complete the survey, so this may be less of a limitation and more a reflection of dog caregiving as historically women's work [68]. Data were owner-reported; while this was essential for data regarding owners' relationships with their dogs, measures of dog behaviour and owner actions may be less reliable due to the known unreliability of owner interpretation and reporting of

dog behaviour [69]. However, this epidemiological approach facilitated collection of a large-scale dataset that practical measures would preclude.

Very few respondents owned Pugs, English Bulldogs or French Bulldogs and thus any effects of this demographic on MDORS could not be effectively explored. Data on education and socio-economic status were not collected and may influence measures of the owner-dog relationship, and thus will be explored in this cohort in future studies.

MDORS scores were not available for respondents who no longer had their dog (and reported this at 21-months), or for those that took part in the original study in puppyhood but chose not to engage in the longitudinal study and were lost to follow up. Social desirability bias may also have influenced whether and how the survey was completed; for example, owners who had rehomed their dog may have chosen not to participate further to avoid reporting their ownership status. As such, the current results may be positively biased and reflect 'best case scenarios' for national pandemic puppy ownership when dogs reached young adulthood.

Other factors not measured here may have resulted in higher relinquishment levels among the wider population of pandemic puppies not included in this study. It is possible that in a sample of owners with lower socio-economic status (SES) that these Perceived Costs may have resulted in higher levels of relinquishment, based on findings from previous studies [39], and possible link to less secure housing another major reason for relinquishment [52]. Measures of SES are included in future time points of this longitudinal study, in light of the current cost of living crisis in the UK.

An owner-dog relationship is a reflection of a history of interactions between owner and dog [70], and by using a survey it is only reported through one party, the owner. The Closeness scale does report some behaviour of the dog towards the owner, but Costs reports the owners attitude towards their dog. Thus in the present study attitude is biased towards the owners' perception of the owner-dog relationship, in common with many other studies [71]. Owner attitude is a useful measure as owners control many factors relating to welfare of their dog, and validation of MDORS in the UK would benefit future analysis of attitudes to UK owners towards their dogs. A strong relationship is not sufficient to ensure good welfare, so MDORS score should not be taken as a surrogate welfare measure for the dog.

In addition, the cross-sectional nature of the current study could not investigate if the increased burden of dog ownership reflected in the Costs scores will result in future relinquishment and behavioural euthanasia; however, future longitudinal analysis of this cohort has the potential to fill this gap. Furthermore, investigation of the relationships between the owner-dog relationship, behaviour, relinquishment and euthanasia in this Pandemic Puppies cohort as they age could inform if MDORS Costs scores are reduced as behaviour improves.

## Conclusions

Many of the risk factors found to decrease the owner-dog relationship in the Pandemic Puppies cohort are not unique to the COVID-19 pandemic time-period and its atypical circumstances. Instead, they largely relate to problem behaviours, which although amplified in this cohort, are highly prevalent in the wider dog population. Interventions to support these owners with evidence-based behavioural advice, utilising reward-based training could improve their relationship with their dog, and the quality of life on both sides of the dog-human relationship. Lessons can also be learned regarding the importance of well-considered puppy acquisitions, with owners whose decision to purchase a puppy was motivated specifically by the pandemic being more likely to experience weaker owner-dog relationships. Identifying effective strategies to support owners in contemplating the stability of their circumstances at the time of proposed acquisition, and to reflect on whether they are suited for long-term dog

ownership may help prevent ill-considered purchasing that results in weak owner-dog relationships and potential future relinquishment.

## Supporting information

**S1 Appendix. 21-Month survey questions relevant to study.**
(PDF)

**S2 Appendix. Risk factor category frequency distributions and univariable associations with Monash Dog Owner Relationship Scale subscales "Perceived Emotional Closeness" and "Perceived Costs".**
(PDF)

**S3 Appendix. Monash Dog Owner Relationship Scale item frequency and subscale score distributions.**
(PDF)

**S4 Appendix. Full mixed linear regression models for Closeness and Costs, with effect sizes and regression diagnostics.**
(PDF)

## Acknowledgments

We would like to thank the following organisations for their support and assistance in disseminating the survey: the British Veterinary Association (BVA), Royal Society for the Prevention of Cruelty to Animals (RSPCA), The People's Dispensary for Sick Animals (PDSA), The Blue Cross, Battersea Dogs and Cats Home, Agria Pet Insurance, Marc Abraham and the All-Parliamentary Dog Advisory Welfare Group (APDAWG), Pets at Home, Vets4Pets, Viovet, the Animal Behaviour and Training Council (ABTC), the International Partnership for Dogs (IPFD), the Brachycephalic Working Group (BWG), Cariad Hound, Pets4Homes, The Doodle Trust, and many other organisations and individuals. We are also grateful to the The Kennel Club (UK) Breed Health Coordinators and The Kennel Club (UK) Breed Health Coordinators Mentors Group for sharing the survey, and The Kennel Club (UK) Charitable Trust for supporting VetCompass (D.G.O., F.C.D., and C.L.P.).

## Author Contributions

**Conceptualization:** Dan G. O'Neill, Zoe Belshaw, Rowena M. A. Packer.

**Data curation:** Bree L. Merritt, Claire L. Brand, Zoe Belshaw, Rowena M. A. Packer.

**Formal analysis:** Bree L. Merritt.

**Funding acquisition:** Zoe Belshaw, Rowena M. A. Packer.

**Investigation:** Claire L. Brand, Zoe Belshaw, Fiona C. Dale, Camilla L. Pegram, Rowena M. A. Packer.

**Methodology:** Bree L. Merritt, Dan G. O'Neill, Claire L. Brand, Rowena M. A. Packer.

**Project administration:** Claire L. Brand, Rowena M. A. Packer.

**Supervision:** Dan G. O'Neill.

**Visualization:** Bree L. Merritt.

**Writing – original draft:** Bree L. Merritt.

**Writing – review & editing:** Bree L. Merritt, Dan G. O'Neill, Claire L. Brand, Zoe Belshaw, Fiona C. Dale, Camilla L. Pegram, Rowena M. A. Packer.

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
