## [Decision Letter · Decision Letter 0]

4 Sep 2024

PONE-D-24-29934Behaviour and other factors influencing the owner-dog bond as dogs reach early adulthood in the UKPLOS ONE

Dear Dr. Merritt,

Thank you for submitting your manuscript to PLOS ONE. After careful consideration, we feel that it has merit but does not fully meet PLOS ONE’s publication criteria as it currently stands. Therefore, we invite you to submit a revised version of the manuscript that addresses the points raised during the review process. You will find a detailed list of reviewer feedback below.

We look forward to receiving your revised manuscript.

Kind regards,

I Anna S Olsson, Ph.D.

Academic Editor

PLOS ONE

Reviewers' comments:

Reviewer's Responses to Questions

**Comments to the Author**

1. Is the manuscript technically sound, and do the data support the conclusions?

Reviewer #1: Yes

Reviewer #2: Yes

2. Has the statistical analysis been performed appropriately and rigorously? 

Reviewer #1: Yes

Reviewer #2: Yes

3. Have the authors made all data underlying the findings in their manuscript fully available?

Reviewer #1: Yes

Reviewer #2: Yes

4. Is the manuscript presented in an intelligible fashion and written in standard English?

Reviewer #1: Yes

Reviewer #2: Yes

5. Review Comments to the Author

Reviewer #1: Dear authors,

Thank you for this great article which addresses a highly relevant issue - the welfare of our pet-dogs.

Ad 1.: The manuscript is technically sound, with a well-structured methodology that supports the conclusions drawn. The use of the MDORS to measure perceived emotional closeness, interaction and perceived cost appears appropriate and provides reliable results when combined with potential risk factors of the owner-dog bond.

Ad 2.: Statistical analysis is carried out rigorously. The manuscript clearly describes the statistical methods used and the results are presented with appropriate measures of statistical significance and confidence intervals. The choice of statistical methods is well suited to the research questions and the data collected.

Ad 3.: The authors have made all underlying data fully available, as indicated in the data availability statement.

Ad 4.: The manuscript is presented in an understandable manner and is written in standard English. The text is clear, concise and free of major grammatical errors, which facilitates the reader's understanding. The structure of the manuscript follows a logical progression from introduction to methodology, results and discussion, which enhances its readability.

Overall, the manuscript is robust, methodologically sound, and well-presented. It provides valuable insights into the factors influencing the owner-dog bond during early adulthood of dogs, particularly in the context of the COVID-19 pandemic. The findings have significant implications for dog welfare and owner support interventions. One aspect to consider in the limitations section is owner-reported data. Reliance on owner-reported data could introduce bias, as owners may have different perceptions and reporting accuracy. Acknowledging this limitation and discussing potential methods to mitigate it (e.g. using objective measures of dog behaviour) would strengthen the manuscript. Another aspect that could enhance the depth and applicability of the research would be to provide more context on how the findings relate to the existing literature. Whilst the discussion refers to some previous studies, a deeper integration and comparison with a wider range of research would help to situate the study within the wider field.

Reviewer #2: Review „Behaviour and other factors influencing the owner-dog bond as dogs reach early adulthood in the UK”

PONE-D-24-29934

Dear authors,

This manuscript addresses a very important and interesting topic by aiming to identify risk factors associated with a weaker owner-dog bond in UK dogs acquired in 2020. The data provided is highly relevant to various areas of research in the human-animal bond and veterinary sciences.

The methodology and analyses are sound, carefully conducted and clearly described. However, I think that some aspects in the introduction and discussion could be better elaborated to improve the manuscript. I would also make a few changes to the helpful illustrations.

.

Title: After reading the manuscript, I felt that the title was too vague for the manuscript. It might be worthwhile to include “Pandemic dogs”, “risk factors” instead of other factors that the title fits better to your actual investigations.

Introduction:

Line 47: Even though the first sentence makes it clear that emotional relationship in this manuscript is described by owner-dog bond, in my opinion the term is not consistently preserved. I would use either bond or relationship uniformly and consistently in the manuscript.

Line 58: I think it is important to clarify here that it is a matter of convenience euthanasia and would go into this aspect including moral challenges (also in veterinary practice) in more detail.

Line 63-69: As these are important factors for the analysis, these factors should not only be listed, but it would be helpful for the reader if the content would be unpacked. E.g. what is meant by lack of problem behaviour.

Line 72- 76: Same applies here. I think this information needs to be unpacked: What negative long-term effects on behaviour and health? I would provide precise examples and descriptions. Further, explain what is meant by “poor-welfare” source?

Line 84: “impact on dog welfare” This is an often-used term and should be clarified at the very beginning what is meant by this and in what way the dog welfare is impacted. Should be unpacked here and will be of help for the whole manuscript and your arguments.

Methods:

Risk factor variables and Table 1:

This table is a bit hard to follow and it might be a good idea to introduce some of them in a more detailed way in the introduction including your assumption in which direction the factors may impact owner-dog bond.

Line 179-183: Here, or in the introduction I would expect a short explanation of differences training methods. This would be further helpful for the discussion.

Results:

Line: 291: “%” is missing

Figure 2: In general, I really like this figure and it provides a good overview of your results. For publication, a better resolution of the figure is needed. The resolution is currently very poor and hard to read.

Few suggestions:

Line 336-343: As this is a really long heading for a table, you can think about to replaces info (Line 336-343) in footnotes

“No change in care provision” and “Change in care provision” reads a bit irritating and it is not very clear what is meant by this while exploring your figure. Might be better to choose a more precise description that reflects this aspect.

Figure 3: In general, I really like this figure and it provides a good overview of your results. For publication, a better resolution of the figure is needed. The resolution is currently very poor and hard to read.

Line 376-382: As this is a really long heading for a table, you can think about to replaces info (Line 336-343) in footnotes

“<6m to acquire” � I would include “decision” here

Line 399-401: Again, an explanation for reader who are not familiar with certain training techniques, would be of help.

Discussion:

Line 419: “impact” instead of “harm”

Line 421: Again, what is meant by dog welfare? See comment for introduction.

Line 428-434: Methods section?

Line 434-436: Should go in the limitation section

Line 441: I would provide more details for each aspect.

Line 445-448: As this is an ongoing debate, I would expect a bit more reflection on euthanasia based on aggressive behavior.

Line 471-476: I think a huge problem is that most of the people are not aware about the time, money they have to spend and too little responsibility is taken by the owners.

Line 501: Not only ability, but also awareness of their responsibilities.

Line 507: Some countries have already introduced a certificate of competence, so that a dog may only be bought after a two-day course. This is also a ‘hurdle’ and can protect against impulse buying.

Line 512-517: Further explanation: The desire to get a dog may have existed for a long time and it was during this phase (with the hype) that the purchase was made. I think that should be taken into account. The decision to buy is different from the phase before that: Have you wanted a dog for a long time?

Line 523-525: Could also be the feeling that you are doing justice to both: The dog and the work

Line 527-536: A bit weak and lacking argumentation for the discussed results.

I wish the authors good luck with the publication and look forward to the final publication of the highly relevant data.

6. PLOS authors have the option to publish the peer review history of their article (what does this mean?). If published, this will include your full peer review and any attached files.

Reviewer #1: No

Reviewer #2: No

---

## [Author Response · Author response to Decision Letter 0]

24 Oct 2024

Dear Dr Olsson,

Thank you for your comments and those of the reviewers. 

The journal formatting and image requirements have been met in the resubmitted files, and the data will be made available on figshare.

I look forward to hearing your decision.

Best wishes

Bree Merritt

Specific responses to the reviewers are below, copied from the Reponse to Reviewers document:

Dear Reviewers,

Thank you for your valuable feedback on our proposed paper. In response, the introduction and discussion sections have been edited to expand the background for this work and link it to existing publications. Consideration of your points raised has made this a stronger and more readable paper. Please see below for individual responses to reviews.

Reviewer 1

Comment: One aspect to consider in the limitations section is owner-reported data. Reliance on owner-reported data could introduce bias, as owners may have different perceptions and reporting accuracy. Acknowledging this limitation and discussing potential methods to mitigate it (e.g. using objective measures of dog behaviour) would strengthen the manuscript.

Response: For measures of owner-dog relationships (our key outcome measures), as we are capturing these measures from the owners’ perspective, it is vital these reports are direct from the owner. We agree that behavioural observations can be more reliable than owner reports in many circumstances; however, here we aimed to collect a large dataset to facilitate powerful epidemiological risk factor analyses, and thus such practical measures of dog behaviour across a range of contexts would not be feasible in this study design. Similar large scale cohort studies in the UK (e.g. Generation Pup) also rely on owner reports in their peer-reviewed publications. 

We have added the following sentence to address this concern lines 609 – 614):

Data were owner-reported; while this was essential for data regarding owners’ relationships with their dogs, measures of dog behaviour and owner actions may be less reliable due to the known unreliability of owner interpretation and reporting of dog behaviour (69). However, this epidemiological approach facilitated collection of a large-scale dataset that practical measures would preclude.

Comment: Another aspect that could enhance the depth and applicability of the research would be to provide more context on how the findings relate to the existing literature. Whilst the discussion refers to some previous studies, a deeper integration and comparison with a wider range of research would help to situate the study within the wider field.

Response: Thank you for this suggestion. We have added some additional context to the discussion

• Subsection “Behaviour and Perceived Costs”: 

o After the first sentence, we added a sentence describing how commonplace problem behaviour is in this cohort (from sister publication (Brand et al. 2024)) and to demonstrate the scale of this issue internationally and its potential widespread impact on dog owner relationships (lines 470-472). 

Problem behaviour has previously been reported as a risk factor for a reduced owner-dog relationship (18,43), relinquishment (52) and euthanasia (53). 

o More context has been added as to how the results of this current study compare and contrast to previous investigations of behavior and the owner-dog relationship, particularly the type of problems behaviour and the relationship outcomes that were measured (lines 497-511):

A Danish study measuring MDORS (19), also found owner perceived problems with fear were associated with increased Perceived Costs but not Closeness. Aggression or being home alone were not shown to be associated with either outcome, perhaps reflecting the different cultural expectations and management of dogs. 

Other studies of showing the relationship between problem behaviour and the owner-dog relationship have examined the interaction between the owner and the dog (18), or attachment (10,43), and thus have not been able to identify the contrasting effect of behaviour on different aspects of the relationship. The low frequency of relinquishment in this study could be further explained by problem behaviour not being associated with decreased Closeness (and thus the close emotional relationship with the dogs buffers the ‘burden’ of owning them resulting in less relinquishment (10)). However, it has been shown that relinquishing owners do not necessarily have lower emotional attachment to their dogs (39) so more information on the changes to the owner-dog relationship over time in relation to risk factors and relinquishment is warranted. 

• Subsection “Training”: Added a comment on the place of these results supporting reward-based training and their importance given the prevalence of use of aversive methods (lines 537-540):

The observation of a weaker owner-dog relationship with increasing use of aversive techniques in this current study supports current evidence promoting reward-based training (21,22,60). This is an important result given the high prevalence of the use of aversive dog training techniques among owners in this (31) and other study populations (21,60,61).

• Subsection COVID-19: added speculation on the challenges of educating owners as to the responsibilities of dog ownership, and how this has been investigated or legislated for elsewhere (lines 558-570):

Studies report that owners’ awareness of the legal responsibilities that come with dog ownership are poor, including those related to safeguarding health and welfare. For example, Irish dog owners were no more knowledgeable than non-dog owners regarding the responsibilities of dog owners prescribed by law in Ireland (64). Consequently, greater understanding is needed of why owners continue to impulsively acquire dogs in the face of such widespread messaging, when the owners’ future circumstances are liable to lead to challenges providing care; and ultimately even relinquishment. Restrictions to current acquisition practices (e.g., compulsory courses that new owners must attend, as already implemented in countries such as Spain since 2023 (65) and in the Canton of Zurich since 2022 (66)) could be an effective deterrent for impulsive acquisition. However, the efficacy of these rules and whether there would be political will and resources to implement them in the UK is currently unknown.

Reviewer 2

Title: After reading the manuscript, I felt that the title was too vague for the manuscript. It might be worthwhile to include “Pandemic dogs”, “risk factors” instead of other factors that the title fits better to your actual investigations.

Thank you. The title has been edited to reflect that the investigations identified risk factors, and that the dogs were pandemic puppies.

New long title: 

Lasting Lockdown Love? Problem behaviour and pandemic and non-pandemic related risk factors influencing the owner-dog relationship in a UK cohort of dogs reaching early adulthood

Introduction:

Line 47: Even though the first sentence makes it clear that emotional relationship in this manuscript is described by owner-dog bond, in my opinion the term is not consistently preserved. I would use either bond or relationship uniformly and consistently in the manuscript.

We agree that consistency is important, and therefore we have edited the manuscript to consistently use the word relationship throughout.

Line 58: I think it is important to clarify here that it is a matter of convenience euthanasia and would go into this aspect including moral challenges (also in veterinary practice) in more detail.

A short overview of circumstances that my lead to, and consequences of, euthanasia have been added. The moral challenges of euthanasia, and whether it is true euthanasia for the benefit of the dog being euthanised, or the death of the dog is chosen as the best outcome for the owner is a complex topic. Giving the topic it’s full due is outside the scope of this paper but agree that more explanation would benefit readers who may be outside of the sheltering or veterinary worlds.

Text added to introduction (lines 57-65):

Relinquishment can harm welfare when there is reduced capacity to care for relinquished dogs in the new temporary or permanent home. Inability to provide for dogs can lead to the morally challenging decision as to whether euthanasia is chosen, either at the level of the owner who no longer feels able to care for the dog or those providing management of unowned dog populations and local authority or non-governmental sheltering organisations (11). In addition to the impact on canine welfare, the ethically complex decision to euthanise a dog for the lack of appropriate care options, however humane the physical act, can cause moral injury to all people involved (12). Euthanasia is also an essential welfare tool, enabling the relief of suffering, and may be withheld to prolong a human’s relationship with their dog. This is especially concerning given the strong owner-dog relationship to dogs with extreme body type (13) that may necessitate euthanasia. 

Line 63-69: As these are important factors for the analysis, these factors should not only be listed, but it would be helpful for the reader if the content would be unpacked. E.g. what is meant by lack of problem behaviour.

We have clarified what is meant by problem behaviour, and explained the expected association between some risk factors and the owner-dog relationship (lines 79 -88):

Dog characteristics associated with a stronger owner-dog relationship include smaller size (18), purebred breed status (compared to crossbreeds) (15), and brachycephalic conformation (with strong owner-dog relationships identified in some extreme brachycephalic breeds: the Pug, French Bulldog, and English Bulldog) (13). In contrast, weaker owner-dog relationships were observed when the only purpose for the dog was companionship compared to working, sporting or show purposes (19). The presence of dog behaviours normally considered a problem by the owner has been associated with a weaker owner-dog relationship (18,20). Reward based, rather than aversive, training techniques to address or prevent problem behaviours, can be more effective (21) and can be associated with a stronger owner-dog relationship (22,23). 

Line 72- 76: Same applies here. I think this information needs to be unpacked: What negative long-term effects on behaviour and health? I would provide precise examples and descriptions. Further, explain what is meant by “poor-welfare” source?

We have explained this phrase used to describe suppliers whose dogs have poor welfare, and have given an example of negative long-term effects (lines 97-101):

They were also more likely to have risked negative long-term effects on behaviour and health by acquiring their puppy from a poor-welfare source, that is, a breeder or third party supplier who may be selling puppies without due provision for their and their parents’ physical, behavioural, and developmental needs and concealing maternal hereditable and infectious ill health (28,29). 

Line 84: “impact on dog welfare” This is an often-used term and should be clarified at the very beginning what is meant by this and in what way the dog welfare is impacted. Should be unpacked here and will be of help for the whole manuscript and your arguments.

A very good suggestion and the introduction has been rewritten accordingly (lines 47-51):

A person’s welfare is their health and happiness (1), and similarly animal welfare has been defined as how the animal is feeling (2). Animal welfare is often interpreted using the Five Domains Model, considering the animal’s mental state regarding their experiences of nutrition, their environment, physical health, and behavioural interactions (3).

Methods:

Risk factor variables and Table 1:

This table is a bit hard to follow and it might be a good idea to introduce some of them in a more detailed way in the introduction including your assumption in which direction the factors may impact owner-dog bond.

We have now introduced the types of risk factors investigated in this study in the introduction, with explanation as to why they are included and the expected effect on the owner-dog relationship. As that has now been covered, the main purpose of the table is now to list the risk factors that were investigated, and have classified into groups of related concepts to aid the reader, alongside a comment on the primary reason for considering that risk factor.

Line 179-183: Here, or in the introduction I would expect a short explanation of differences training methods. This would be further helpful for the discussion.

An explanation of training method expected association with relationship has been added to the introduction, addressed above. In the methods, more detail has been provided on how (Brand et al. 2024) derived a training style from reported use of training methods (line 218-224).

Respondents reported training methods or aids that they used on their dog up to 21-months of age. Based on the principles of operant conditioning, each method or aid was classified was as reward-based (positive reinforcement, negative punishment) or aversive (negative reinforcement, positive punishment). Respondents were allocated a training style based on the classification of methods or aids they used as reward-only, aversive-only, rewards with one aversive method, or rewards with more than one aversive method (31).

Results:

Line: 291: “%” is missing

Added.

Figure 2: In general, I really like this figure and it provides a good overview of your results. For publication, a better resolution of the figure is needed. The resolution is currently very poor and hard to read.

This figure has been revised and resolution improved.

Few suggestions:

Line 336-343: As this is a really long heading for a table, you can think about to replaces info (Line 336-343) in footnotes

We have separated it into the figure title (in bold) and the figure legend (normal text) as per the PLOS formatting guidelines.

“No change in care provision” and “Change in care provision” reads a bit irritating and it is not very clear what is meant by this while exploring your figure. Might be better to choose a more precise description that reflects this aspect

We have changed this to “change in care arrangements” to be clearer.

Figure 3: In general, I really like this figure and it provides a good overview of your results. For publication, a better resolution of the figure is needed. The resolution is currently very poor and hard to read.

This figure has been revised and resolution improved.

Line 376-382: As this is a really long heading for a table, you can think about to replaces info (Line 336-343) in footnotes

We have separated it into the figure title (in bold) and the figure legend (normal text) as per the PLOS formatting guidelines.

“<6m to acquire” � I would include “decision” here

Decision is now included.

Line 399-401: Again, an explanation for reader who are not familiar with certain training techniques, would be of help.

Further explanation added to introduction and methods. This paragraph edited for clarity (lines .

Using only one aversive-based method compared to two or more aversive-based training methods was associated with increased Closeness scores (Fig 3). 

Discussion:

Line 419: “impact” instead of “harm”

Changed as suggested.

Line 421: Again, what is meant by dog welfare? See comment for introduction.

Added to introduction as suggested.

Line 428-434: Methods section?

This repetition of the reliability result was removed.

Line 434-436: Should go in the limitation section

Moved to limitations as suggested.

Line 441: I would provide more details for each aspect.

This has now been elaborated more in the introduction as described in the above responses.

Line 445-448: As this is an ongoing debate, I would expect a bit more reflection on euthanasia based on aggressive behavior.

We agree that euthanasia based on aggressive behaviour is an important and topical point. However, there was an insufficient sample to specifically analyse this outcome in this study and thus we had not focused upon this heavily in the discussion. We have further reflected on your point and added a suggestion for future research below based on your sug

---

## [Decision Letter · Decision Letter 1]

9 Dec 2024

Lasting Lockdown Love? Problem behaviour and pandemic and non-pandemic related risk factors influencing the owner-dog relationship in a UK cohort of dogs reaching early adulthood

PONE-D-24-29934R1

Dear Dr. Rowena M A Packer

We’re pleased to inform you that your manuscript has been judged scientifically suitable for publication and will be formally accepted for publication once it meets all outstanding technical requirements.

Kind regards,

Joshua Kamani, PhD

Academic Editor

PLOS ONE

Additional Editor Comments (optional):

Reviewers' comments:

Reviewer's Responses to Questions

**Comments to the Author**

1. If the authors have adequately addressed your comments raised in a previous round of review and you feel that this manuscript is now acceptable for publication, you may indicate that here to bypass the “Comments to the Author” section, enter your conflict of interest statement in the “Confidential to Editor” section, and submit your "Accept" recommendation.

Reviewer #2: All comments have been addressed

2. Is the manuscript technically sound, and do the data support the conclusions?

Reviewer #2: (No Response)

3. Has the statistical analysis been performed appropriately and rigorously? 

Reviewer #2: Yes

4. Have the authors made all data underlying the findings in their manuscript fully available?

Reviewer #2: Yes

5. Is the manuscript presented in an intelligible fashion and written in standard English?

Reviewer #2: Yes

6. Review Comments to the Author

Reviewer #2: (No Response)

7. PLOS authors have the option to publish the peer review history of their article (what does this mean?). If published, this will include your full peer review and any attached files.

Reviewer #2: **Yes: **Svenja Springer

---

## [Editor Report · Acceptance letter]

10 Jan 2025

PONE-D-24-29934R1 

PLOS ONE

Dear Dr. Packer, 

I'm pleased to inform you that your manuscript has been deemed suitable for publication in PLOS ONE. Congratulations! Your manuscript is now being handed over to our production team.

Kind regards, 

on behalf of

Dr. Joshua Kamani 

Academic Editor

PLOS ONE